# Assessing the Quality of YouTube’s Incontinence Information after Cancer Surgery: An Innovative Graphical Analysis

**DOI:** 10.3390/healthcare12020243

**Published:** 2024-01-18

**Authors:** Alvaro Manuel Rodriguez-Rodriguez, Marta De la Fuente-Costa, Mario Escalera-de la Riva, Fernando Domínguez-Navarro, Borja Perez-Dominguez, Gustavo Paseiro-Ares, Jose Casaña-Granell, María Blanco-Diaz

**Affiliations:** 1Physiotherapy and Translational Research Group (FINTRA-RG), Institute of Health Research of the Principality of Asturias (ISPA), University of Oviedo, 33003 Oviedo, Spain or uo22174@uniovi.es (A.M.R.-R.); fuentemarta@uniovi.es (M.D.l.F.-C.); escaleramario@uniovi.es (M.E.-d.l.R.); blancomaria@uniovi.es (M.B.-D.); 2Faculty of Medicine and Health Sciences, University of Oviedo, 33006 Oviedo, Spain; 3Exercise Intervention for Health Research Group (EXINH-RG), Department of Physiotherapy, University of Valencia, 46010 Valencia, Spain; fernando.dominguez@uv.es (F.D.-N.); jose.casana@uv.es (J.C.-G.); 4Psychosocial Intervention and Functional Rehabilitation Research Group, Faculty of Physiotherapy, University of A Coruña, 15008 Coruna, Spain; gustavo.paseiro@udc.es

**Keywords:** cancer, incontinence, surgery, information, quality, YouTube, DISCERN, GQS, JAMA, PEMAT, MQ-VET, Sankey, chord, diagram

## Abstract

Background: Prostate and colorectum cancers rank among the most common cancers, and incontinence is a significant postsurgical issue affecting the physical and psychological well-being of cancer survivors. Social media, particularly YouTube, has emerged as a vital source of health information. While YouTube offers valuable content, users must exercise caution due to potential misinformation. Objective: This study aims to assess the quality of publicly available YouTube videos related to incontinence after pelvic cancer surgery. Methods: A search on YouTube related to “Incontinence after cancer surgery” was performed, and 108 videos were analyzed. Multiple quality assessment tools (DISCERN, GQS, JAMA, PEMAT, and MQ-VET) and statistical analyses (descriptive statistics and intercorrelation tests) were used to evaluate the characteristics and popularity, educational value, quality, and reliability of these videos, relying on novel graphical representation techniques such as Sankey and Chord diagrams. Results: Strong positive correlations were found among quality rating scales, emphasizing agreement. The performed graphical analysis reinforced the reliability and validity of quality assessments. Conclusions: This study found strong correlations among five quality scales, suggesting their effectiveness in assessing health information quality. The evaluation of YouTube videos consistently revealed “high” quality content. Considering the source is mandatory when assessing quality, healthcare and academic institutions are reliable sources. Caution is advised with ad-containing videos. Future research should focus on policy improvements and tools to aid patients in finding high-quality health content.

## 1. Introduction

Prostate and colorectum cancers rank as the second and third most common cancers among males, superseded only by lung cancer, as per the International Agency for Research on Cancer, an organizational body of the World Health Organization (WHO) [1]. For females, colorectal cancers (encompassing malignancies of the colon, rectum, anus, etc.) constitute the second most prevalent type after breast cancer, according to the same agency [2]. While therapeutic advancements in radiation, chemotherapy, immunotherapy, and targeted modalities have broadened treatment options across numerous cancer types, surgical resection persists as the definitive primary intervention for most bladder, prostate, cervical, uterine, rectal, and anal tumors [3]. Pelvic cancer surgeries such as radical prostatectomy, cystectomy, hysterectomy, and abdominoperineal resection necessitate extensive training and finesse to balance oncologic outcomes with quality of life (QoL) factors, including continence and sexual function. Improvements in minimally invasive and robotic-assisted surgical platforms are enhancing access and precision for pelvic cancer resections while aiming to mitigate collateral impairment and morbidity. However, even with technological progress, surgical expertise and experience remain paramount for optimizing outcomes in pelvic oncology [4].

One potential postoperative side effect of cancer surgery is incontinence, especially in patients undergoing interventions for the aforementioned cancer types [5]. Postoperative incontinence is a prevalent concern among patients undergoing certain cancer treatments, such as rectal cancer management or prostate cancer surgery [6]. It refers to the loss of control over urinary or fecal functions, resulting in unintentional leakage or inability to regulate urination or bowel movements [7,8].

Incontinence represents a distressingly common complication that profoundly impacts the quality of life for numerous cancer survivors after invasive pelvic operations [9]. However, despite significant advancements in surgical techniques over recent decades, postoperative incontinence remains a challenging problem that affects patients physically and psychologically. Reported prevalence is highly variable based on cancer type, surgical method, and outcome measurement, with urinary incontinence rates ranging from 5% to 70% (most studies 25–45%) and fecal incontinence between 9% and 68% [10], increasing with age [11]. The resulting loss of urinary or bowel control can have profound emotional and practical impacts, leading to social seclusion, sexual intimacy struggles, embarrassment, diminished self-esteem, and reduced overall QoL [12,13]. Coping with constant leakage and involuntary loss of basic bodily function control presents unique challenges for cancer survivors’ psychological well-being and sexual function, among other domains [14,15].

In recent years, the advent of social media has transformed how individuals seek and share health-related information. Cancer surgery survivors are no exception, increasingly turning to e-health [16] and social networks [17]. YouTube garners 95.0% of global internet users, making it the largest video-sharing platform worldwide. With over 122 million daily users, its colossal user base solidifies its status as the preeminent video website [18]. YouTube’s influence is evident through its 122 million daily active users and staggering 1 billion hours of content watched daily [19]. Thus, YouTube has become one of the most prominent platforms for patients to access information regarding their conditions [20]. As one of the most utilized social networks globally and the largest video website, YouTube provides a vast repository of user-generated content related to cancer surgery and its aftermath [21,22].

YouTube’s accessibility and user-friendly interface make it a preferred destination for cancer surgery survivors seeking information about their specific conditions, treatment options, potential side effects, and coping strategies for postoperative challenges [23]. With a simple search, survivors can find abundant videos uploaded by healthcare professionals, patient advocates, fellow survivors, and others sharing experiences and insights. YouTube videos’ visual and audio formats allow survivors to connect emotionally with the content, enabling a deeper understanding of the challenges and triumphs faced by others undergoing similar procedures [24,25].

Engaging in online discussions, joining patient support groups, and participating in virtual events on YouTube enables cancer surgery survivors to find community and support that may be less accessible offline [26]. By interacting through comments and messages, survivors can seek advice, share experiences, and learn from diverse perspectives. The platform fosters a culture of knowledge exchange and empowerment, providing cancer survivors validation, encouragement, and solidarity during the post-treatment recovery process [27].

In navigating the extensive landscape of YouTube for health-related content post cancer surgery, survivors must exercise caution and critical thinking due to the potential prevalence of misinformation and unsubstantiated claims [28]. As highlighted by studies on the ‘Dr. Google’s effect’, YouTube serves as a vast repository where misinformation and disinformation can freely circulate, posing risks of confusion and inaccurate information [29,30]. To address this, healthcare professionals play a crucial role in guiding survivors toward credible YouTube channels and promoting fact-checking against reputable medical sources [29]. By acknowledging the interdisciplinary nature of YouTube as a platform for health information and leveraging evidence-based practices, healthcare providers can enhance digital health literacy and empower survivors to make informed decisions regarding their cancer care [31,32].

As the internet has become a common health information source for many patients, concerns exist about the variable quality of uncontrolled online content from platforms like YouTube. Studies indicate that most patients access online health information independently without healthcare professional input [22]. While the internet can fill knowledge gaps, patients often lack the skills to appraise source credibility.

This research aims to examine the quality and accuracy of online health information patients access through media-sharing platforms regarding a highly prevalent pathology in both genders. Using five validated quality assessment tools (e.g., DISCERN, GQS, JAMA, PEMAT, MQ-VET), the study analyzes a sample of publicly available YouTube videos on this health topic to evaluate their educational value and reliability. The goal is to provide insights into the usefulness of this prevalent yet uncontrolled online health media, given its widespread use by patients seeking knowledge. By assessing the quality of web-based resources patients often access independently, this study aims to inform efforts guiding patients and professionals toward trustworthy online health information to enhance engagement in e-health. Additionally, the importance of this research lies in addressing the evolving landscape of postoperative care, where online platforms play a crucial role in shaping patient experiences and decisions, ultimately contributing to the broader discourse on patient education, empowerment, and healthcare outcomes.

## 2. Materials and Methods

On 22 March 2023, a search was conducted on the website http://www.youtube.com with the search terms “*Incontinence after cancer surgery*”. An initial cohort of 160 videos was selected from the available options, to reproduce a simple search approach executable by any person. YouTube ranked the video results based on relevance using their proprietary algorithm on the specified date. All videos were then compiled into a spreadsheet and screened for duplicates. The research team applied the exclusion criteria, eliminating non-English language videos, duplicates, advertisements, and content unrelated to incontinence. Finally, 108 videos were assigned to two independent examiners who viewed, analyzed, and evaluated them over a three-week period, as depicted in the flow diagram adhering to PRISMA guidelines [33,34] (see Figure 1).

### 2.1. Statistical Analysis

#### 2.1.1. Outcome Measures

Each video’s descriptive details were compiled, including the upload date and days online, length in minutes and seconds, number of views (view counts), likes, comments, and subscribers. Videos were categorized by continent of origin based on additional characteristics: Africa, America, Asia, Australia, and Europe. Academic institutions, media (newspapers, TV, etc.), health institutions, nongovernmental organizations (NGOs), healthcare individuals (healthcare workers, etc.), and non-healthcare individuals (other people, usually patients, explaining their experiences in this matter) were the six categories that the videos were divided into.

In addition, information on the gender of the film’s target audience (male, female, both) and whether the movie considered pharmacological therapy (Yes/No) was collected. This information was also gathered based on the type of incontinence (urinal incontinence, fecal incontinence, or both). The cancer type specified in the data (prostate, uterus, bladder, colorectal, other) was also acquired. Finally, information about the type of treatment intervention mentioned in the movie was acquired (physiotherapy, surgery, physiotherapy and surgery, drugs or orthotics, education, or others). View ratio (*VR*) [35,36] (see Equation (1), View ratio equation) was used to determine video popularity:(1)$$VR=\frac{View\;Count}{Days\;Online}$$

The reliability and instructional quality of the 108 included movies were evaluated using the DISCERN scale [37,38] and the Global Quality Scale (GQS) [26]. The Journal of the American Medical Association (JAMA) [39] created benchmark standards for evaluating video accuracy and reliability. The Patient Education Materials Assessment Tool (PEMAT) [40] was used to assess the understandability and actionability of patient education materials. The Medical Quality Video Evaluation Tool (MQ-VET) was employed to gauge the overall quality of online medical videos [41].

#### 2.1.2. DISCERN Tool

In this study, a modified 5-point version of the original DISCERN tool was utilized to evaluate the reliability and educational quality of the records. This adapted instrument comprises five distinct questions, assigning one point if the video satisfies the respective criterion and zero points if not. The original questionnaire, termed “Quality Criteria for Consumer Health Information”, was initially developed by the University of Oxford’s Public Health and Primary Care Division (London, UK) to assess information quality related to treatment options for health conditions. It was first published in 1999 [37]. DISCERN scores ranging from 4 to 5 are categorized as “Very High”, 3 to 4 as “High”, 2 to 3 as “Average”, 1 to 2 as “Low”, and 0 to 1 as “Very Low”. Higher scores indicate greater information quality [38].

#### 2.1.3. Global Quality Score (GQS) Tool

The GQS evaluates five factors in each video to determine the overall quality of online content. Each of the five recognizable criteria present in a movie is allocated one point, similar to the DISCERN tool, with a maximum educational quality score of 5 points [42] This scale considers factors such as information accessibility and quality, overall information flow, and the utility of content to users [26].

#### 2.1.4. JAMA Tool

The video accuracy and reliability were evaluated using the benchmark standards developed by the Journal of the American Medical Association (JAMA), which is a four-criteria (authorship, attribution, disclosure, and currency) assessment tool [39]. The authorship item emphasizes the importance of correctly reporting the author’s name and affiliation on the website. The attribution item assesses the effectiveness of citing the website’s content. The currency item determines whether the website includes the uploading and updating dates. Finally, the disclosure section discloses whether the site owner has identified any potential conflicts of interest. Each item is scored by examiners on a range of 1 to 4 points [22,43].

#### 2.1.5. Patient Education Materials Assessment Tool (PEMAT)

The Agency for Healthcare Research and Quality (AHRQ) developed a reliable and valid tool called PEMAT [40] to evaluate the understandability and actionability of patient education materials. Understandability refers to the capacity of individuals from diverse backgrounds to comprehend and explain key messages. Actionability assesses the degree to which individuals with varying health literacy and backgrounds can determine appropriate actions to improve their health based on the information provided. PEMAT consists of two sections: PEMAT-P for printable materials like brochures and PDFs and PEMAT-A/V for audiovisual content such as videos and multimedia, the latter being the version utilized here. Scores are calculated by summing points and then dividing by the total possible points; maximum scores are 1.0 for understandability and 1.0 for actionability. The total maximum score is the sum of these scores, or 2.0. The developers established a cutoff of 0.7 for both understandability and actionability [44,45].

#### 2.1.6. Medical Quality Video Evaluation Tool (MQ-VET)

The MQ-MET is a validated questionnaire that assesses the quality of internet medical videos provided by both doctors and members of the general public [41]. This technique is likely to play a critical role in the future establishment of standardized evaluations of online medical films. The MQ-VET tool’s final version consists of four portions that examine distinct aspects, each with a varied number of questions. Part 1 consists of 5 questions, Part 2 of 4 questions, Part 3 of 3 questions, and Part 4 of 3 questions, for a total of 15 questions for the complete tool. All of these questions are graded on a Likert scale of 1 to 5 points (with 1 point for “Strongly Disagree” and 5 points for “Strongly Agree”). Finally, the total score is the sum of all the question scores, with a maximum score of 75 points [41].

### 2.2. Statistical Assessment

Two statistical analyses were conducted on the dataset. First, descriptive statistics were analyzed, and correlational relationships between variables were explored using Pearson correlation coefficients [46,47]. To visually represent these associations, two innovative data visualization techniques were employed to enhance reader comprehension [48] of the flow between quality questionnaires: Sankey and chord diagrams. These diagrams require programming expertise [49] but effectively convey multidimensional data interrelationships [50].

Sankey diagrams are a type of flow chart originally developed in 1896 by Irish Captain Matthew Henry Phineas Sankey to visualize steam engine energy efficiency [51]. They are also popular in physics, economics, and business to examine complex multistep processes [52]. As available data become increasingly granular, Sankey diagrams have become a prevalent visualization for system flows and transfers [50]. Despite limited use thus far in medical research, Sankey diagrams are gaining traction for their lucid representation of information flow [53], particularly in health services research and public health to depict patient navigation of healthcare systems [54] and other big healthcare data applications [55]. Recent studies also apply Sankey diagrams in cancer research [52,56,57].

In the chord diagram, circular visualizations portray relationships among system entities. States are depicted as arcs, and connections between them are illustrated as chords. This visual tool, although less common in scientific literature [58], proves valuable for intuitively grasping aggregated statistics [59] and connections within clustered elements [59]. Each link in the diagram signifies sample transitions from a source to a target subgroup, with wider links indicating more frequent transitions [60]. Unlike physical flows, the widths of these links convey informational relationships. Nodes within the diagram represent categories, not discrete processes [50].

## 3. Results

An interclass correlation coefficient (ICC) analysis was performed to measure examiner agreement utilizing the mean rating (k = 2), consistency, two-way random model, and Pearson’s correlation technique. The 95% confidence intervals (CI) were included in the analysis. The level of significance was set at *p* < 0.05, and the values produced from the ICC calculation can be classified as follows for this level: values less than 0.5 denote “Poor” reliability; values between 0.5 and 0.75 denote “Moderate” reliability; values between 0.75 and 0.90 denote “Good” reliability; and values greater than 0.90 denote “Excellent” reliability [60]. This study has an inter-rater agreement score of 0.9465.

### 3.1. Descriptive Statistics

Descriptive statistics were calculated for each video to establish the mean, standard deviation (SD), median, minimum, and maximum values. The quality scales were subjected to the same computations. This information can be seen in Table 1.

Table 2 shows the scores in the DISCERN, GQS, JAMA, PEMAT, and MQ-VET tools according to the origin of the video, the author (or uploader), the gender, and the cancer type, expressed in mean and SD for each of them.

Regarding the source of production (author or uploader of the films), 35.18% were uploaded by health individuals, followed by health institutions (34.25%), academic institutions (12.03%), non-health individuals (8.33%), NGOs (6.48%), and media (3.70%). Table 3 contains these results.

Stratified by target patient gender, 89.81% of videos were intended for male cancer patients and 10.19% for females (Table 4). Among female-oriented videos, the highest view percentages addressed uterine (4.63%) and other cancer types (4.63%), with a small proportion of bladder cancer (0.93%). No colorectal cancer videos for females were identified. In contrast, prostate cancer predominated male-focused videos at 82.41% of cases. Bladder cancer ranked as the second most prevalent, accounting for 7.41% of all cases, while colorectal and other types of cancer had no videos for males.

### 3.2. Correlation within Quality Scales

Table 5 displays the Pearson correlation coefficients and 95% confidence interval limits for the quality scales in relation to each other. All of them show statistically significant relationships (*p* < 0.01).

These correlation coefficient data can be graphically understood in the Sankey and chord diagrams below. The chord diagram of the scores received by the videos categorized in each of the scales examined is shown in Figure 2. The groups on the periphery display movies with scores of high quality (HQ) and low quality (LQ) on each of the scales analyzed in this study. The band’s widths that connect the groups reflect the number of videos that receive the score indicated in the name of the two groups that each band connects.

For further clarification, Figure 3 depicts the 51 videos rated as high quality (HQ) on the MQ-VET scale (Table 6) that also achieved high-quality scores on DISCERN (48 videos), GQS (47 videos), JAMA (49 videos), and PEMAT (51 videos, i.e., all). However, only 3, 4, and 3 of these MQ-VET HQ videos received low-quality (LQ) scores on the DISCERN, GQS, and JAMA scales, respectively.

Figure 4 shows the entire Sankey diagram, with the quality scales examined on the left (DISCERN, GQS, MQ-VET, JAMA, and PEMAT), followed by an “HQ” for high-quality videos grouped according to that scale or an “LQ” for low-quality movies grouped according to that scale. The two primary clusters on the right of the diagram are used to classify videos as high (HQ) or low (LQ) quality based on their total quality (TQ) score.

In Figure 5, we concentrate on the high total quality videos (TQ HQ), and we can see that almost all of the videos that received high-quality scores in the previous scales make up the final grouping of high total quality videos, with the exception of one video that received low quality per the GQS and MQ-VET scales—that, in the aggregate evaluation of all the scales, is classified as high total quality.

Similarly, in Figure 6, the analysis of the low total quality videos (TQ LQ) shows that, with the exception of a few videos that, like in the previous instance, are not representative of the other vast majority, the vast majority of videos that received a low aggregate quality score came from receiving a low-quality rating on the scales compared.

## 4. Discussion

The primary aim of this research was to gain a comprehensive understanding of the quality of information that patients independently access via a prominent online media-sharing platform. In recent years, the internet has emerged as a pivotal source of health-related information for both the general public and healthcare practitioners, largely attributable to the widespread adoption of modern technology and the extensive availability of content. Nonetheless, the unrestricted accessibility of online material carries inherent risks. Consequently, individuals and healthcare consumers may encounter misleading or potentially harmful information. Emphasizing the importance of robust education and dissemination, ideally overseen by healthcare professionals, becomes imperative for enhancing healthcare protocols.

Among the scrutinized videos, a majority originated from health individuals, health institutions, and academic institutions (constituting 35.18%, 34.25%, and 12.03%, respectively), affording a certain degree of reliability given their high standing on quality assessments. Notably, these three categories consistently achieved the highest quality scores across all evaluated metrics, with academic institutions consistently leading the way. Conversely, non-health individuals contributed a noteworthy proportion of videos (8.33%), yet these videos consistently obtained the lowest quality scores across all metrics without exception.

In our study, the analyzed videos had an average duration of 15 min and 8 s, which closely mirrors the average length reported in prior studies. Specifically, previous research focusing on videos demonstrating pelvic floor exercises following prostatectomy surgery reported a mean length of 14:42 min [20]. Notably, we did not identify a statistically significant correlation between the length of the videos and their quality or popularity. The intraclass correlation coefficient (ICC) serves as a widely used metric for assessing test-retest, intra-rater, and inter-rater reliability. By utilizing the 95% confidence interval of the ICC estimation [60], we determined that the inter-reviewer agreement in our study was exceptionally high at 0.9465, signifying an outstanding level of agreement between the two assessors.

Furthermore, we identified a statistically significant pairwise correlation among the DISCERN, GQS, JAMA, PEMAT, and MQVET scales (all with a *p*-value < 0.01), as illustrated in Table 5. When evaluating the educational quality of the videos using these scales, it is essential to note that, based on the DISCERN scoring system, the average score achieved by the videos falls within the “High” category. According to the DISCERN scale, scores ranging between 4 and 5 points are classified as “Very High”, 3 to 4 as “High”, 2 to 3 as “Average”, 1 to 2 as “Low”, and “Very Low” between 0 and 1. Higher scores on the scale indicate a higher level of information quality [36].

If we normalize all the scales within the 0 to 5 range (see Table 7), it becomes evident that the overall average score obtained by the dataset within each scale consistently falls within the 3 to 4 range, with minimal disparities. This robustly supports our statistical analysis, demonstrating a significant correlation across the five scales applied to the dataset.

In this health research study, we conducted Pearson’s correlation analysis to examine the relationships among the quality rating scales, which include DISCERN, GQS, JAMA, PEMAT, and MQ-VET. As previously mentioned, our rigorous statistical analysis revealed robust, highly significant positive correlations (with *p*-values < 0.01) among these scales, indicating a remarkable level of consistency and convergence across the various evaluation criteria. To further emphasize and visually represent these strong interconnections, we employed chord diagrams. Within these diagrams, it was evident that samples receiving high ratings on one quality scale also tended to receive high ratings on the other scales, reinforcing the coherence in our assessments. Conversely, samples with low-quality ratings on one scale consistently exhibited low ratings across all the other scales. These chord diagrams provided a vivid graphical depiction of the pronounced correlations identified in our statistical tests. Notably, only a few samples, as indicated in Table 5, displayed variations in their ratings across different scales, which were not statistically significant.

Collectively, the comprehensive statistical correlation analysis, combined with the visual clarity offered by the chord diagrams, underscored the substantial interrelationships and strong agreement among these quality rating scales. As depicted in Figure 7, the chord diagrams for high-quality (HQ) scores on the quality scales (Figure 7A,C,E,G,I) featured wide bands, signifying that the majority of videos receiving high ratings on one scale also received high ratings on the other scales. Conversely, chord diagrams for low-quality (LQ) scores (Figure 7B,D,F,H,J) exhibited broad bands with low-quality scores on the other scales and, in some cases, displayed thin bands or no bands at all with high-quality scores. This visual representation further underscores the consensus and alignment among the different quality rating scales employed in our study.

### 4.1. Limitations

In alignment with user experience simulation principles, we refrained from conducting searches in incognito mode to mitigate the potential influence of browsing history and geographical locations. It is essential to note that while the content on YouTube undergoes continuous evolution, our analysis accurately captures the state of the videos at a specific moment in time [20].

For the purposes of this study, we exclusively considered videos that were directly accessed on YouTube through the search query “Incontinence after cancer surgery”. We deliberately excluded external links originating from other medical-related websites from our analyses.

Given the dynamic nature of YouTube, where new videos are consistently uploaded, viewed, and rated, it is imperative to acknowledge that our search efforts were limited to the first 160 videos. This approach mirrors the typical behavior of the average internet user, as extensive research has elucidated that most individuals do not extend their searches beyond the initial 50 results [61,62].

### 4.2. Future Research

The involvement of academic and health institutions, along with health professionals, government health authorities, and legislators is crucial in formulating policies that enhance the quality of information available on the internet, leading to a positive influence on the health behaviors of the population. Consequently, in future research, it is essential to create and identify tools that assist patients in differentiating the highest quality health-related videos more easily.

## 5. Conclusions

In conclusion, this study focused on independently accessed online health information, revealing a consistently high level of quality in YouTube videos related to “Incontinence after Cancer Surgery”. Our evaluation, utilizing five quality scales and reinforced by analytical diagrams, emphasizes the robustness of our findings. Notably, healthcare, and academic institutions, as well as healthcare professionals, emerge as reliable sources due to their precise terminology and higher adherence to quality indicators. Patients are advised to prioritize these sources and exercise caution with non-health individuals’ content, especially videos featuring advertisements, which may pose potential conflicts of interest.

## Figures and Tables

**Figure 1 healthcare-12-00243-f001:**
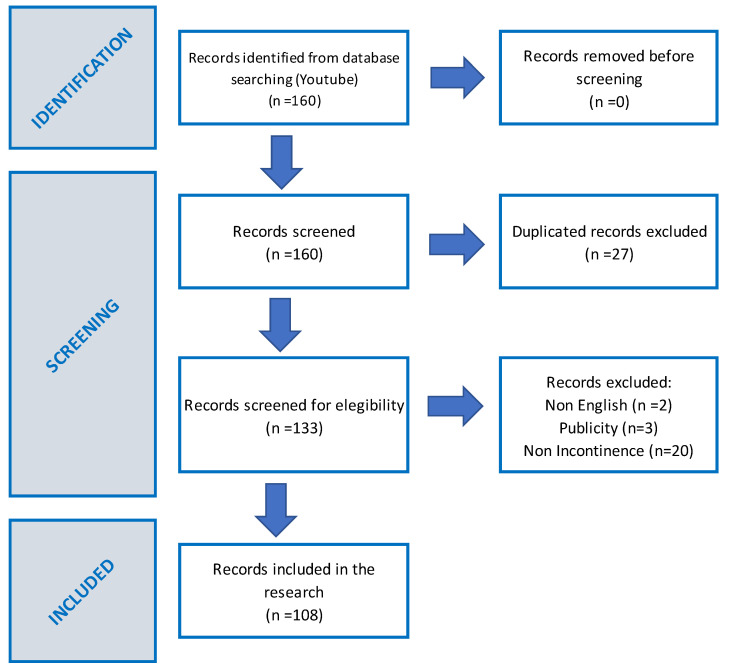
PRISMA flow diagram of records included in the research.

**Figure 2 healthcare-12-00243-f002:**
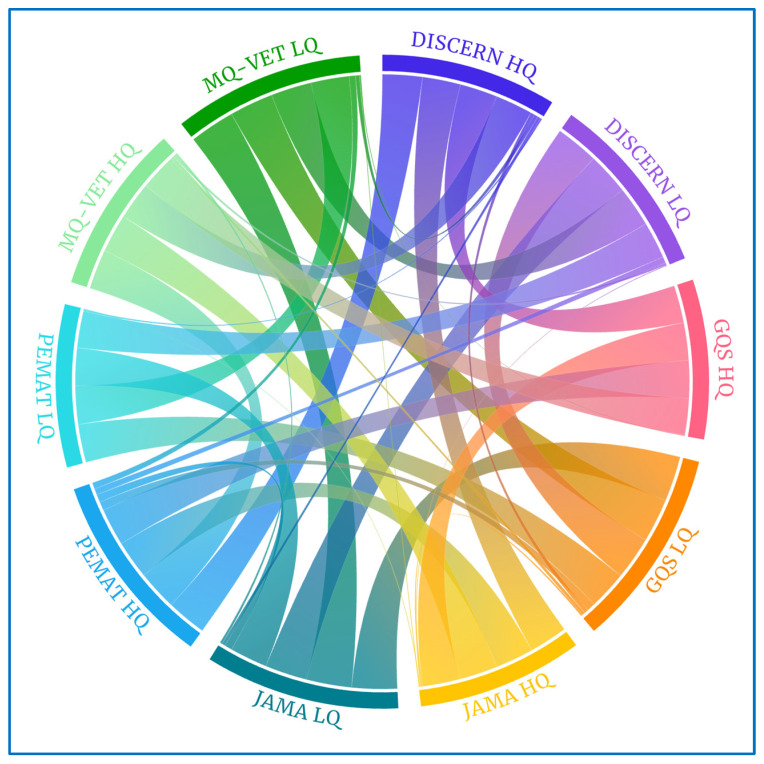
Chord diagram for all quality scales.

**Figure 3 healthcare-12-00243-f003:**
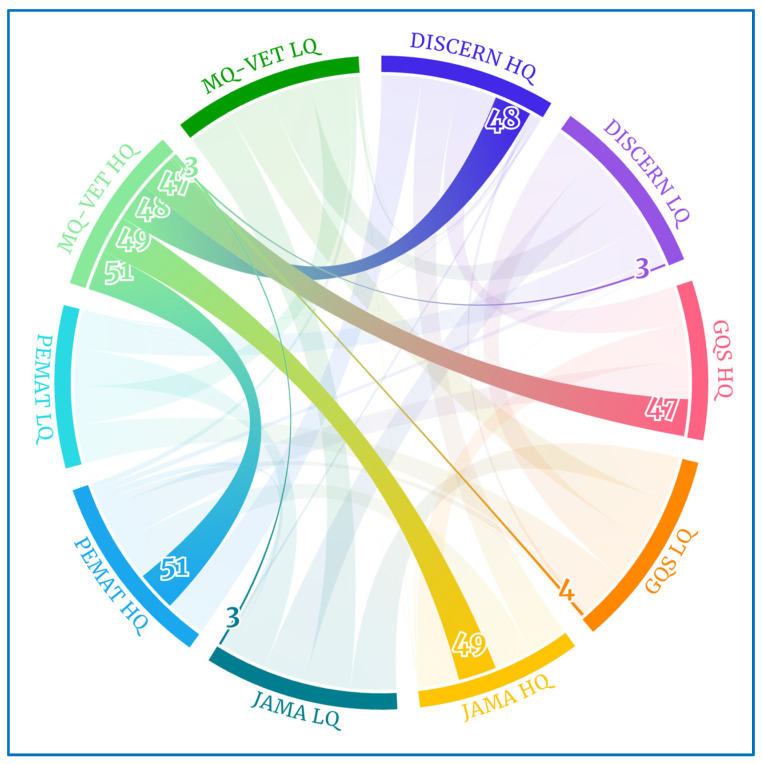
Chord diagram for MQ-VET HQ videos.

**Figure 4 healthcare-12-00243-f004:**
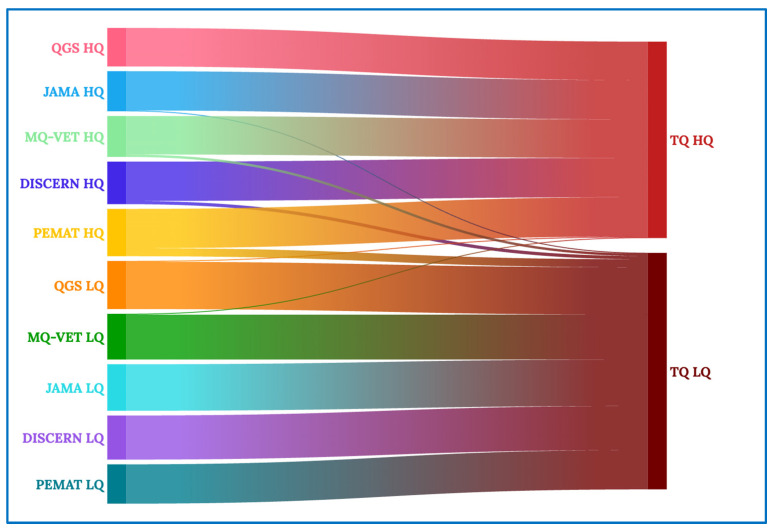
Complete Sankey diagram.

**Figure 5 healthcare-12-00243-f005:**
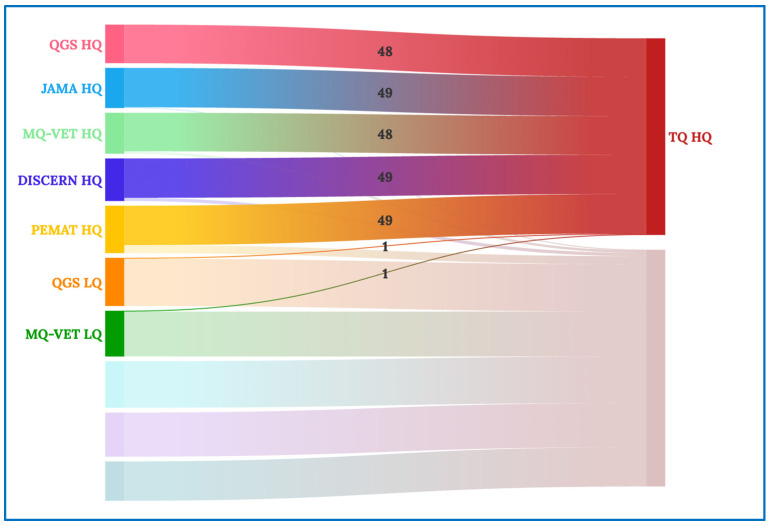
Sankey diagram for high-quality videos.

**Figure 6 healthcare-12-00243-f006:**
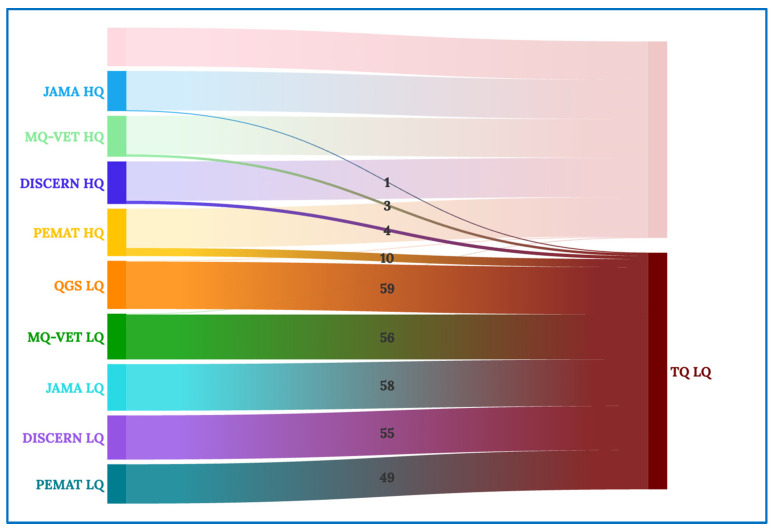
Sankey diagram for low-quality videos.

**Figure 7 healthcare-12-00243-f007:**
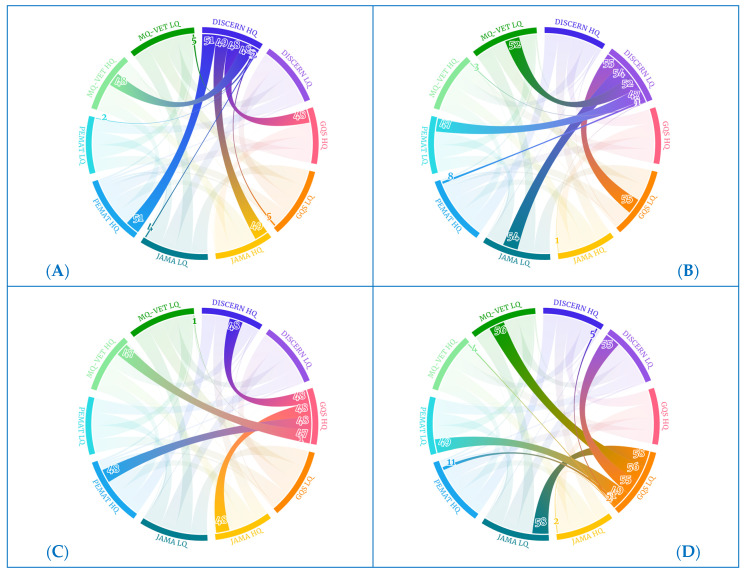
Chord diagrams for each quality scale.

**Table 1 healthcare-12-00243-t001:** Descriptive statistics of the variables recorded for the study.

	Mean	SD	Median	Min	Max
Days online	1921.57	1276.95	1614.50	81.00	5150.00
Duration (hh:mm:ss)	0:15:08	0:19:02	0:06:22	0:00:23	1:08:10
Views (*n*)	79,334.47	582,735.22	3376.00	40.00	6,032,889.00
Likes (*n*)	690.00	5504.28	18.50	0.00	57,174.00
Comments (*n*)	51.83	386.74	0.00	0.00	4018.00
Subscribers (*n*)	129,693.65	590,655.89	4570.00	0.00	5,870,000.00
View ratio (*n*)	49.63	341.47	2.50	0.00	3447.00
DISCERN	3.37	0.92	3.00	1.00	5.00
GQS	3.22	0.89	3.00	1.00	5.00
JAMA	2.48	0.72	2.00	1.00	4.00
PEMAT	1.35	0.29	1.38	0.65	1.92
MQ-VET	10.38	1.87	10.00	6.00	14.00

**Table 2 healthcare-12-00243-t002:** Mean and SD values of the scales according to the origin, author, gender, and cancer type.

	DISCERN		GQS		JAMA		PEMAT		MQ-VET	
	Mean	SD	Mean	SD	Mean	SD	Mean	SD	Mean	SD
**ORIGIN**										
1. Africa	0.00	0.00	0.00	0.00	0.00	0.00	0.00	0.00	0.00	0.00
2. America	3.41	0.90	3.27	0.90	2.52	0.68	1.35	0.29	52.50	9.15
3. Asia	3.00	0.00	2.5	0.70	1.50	0.70	1.14	0.15	42.50	3.50
4. Australia	3.27	1.00	3.18	0.98	2.18	0.75	1.32	0.27	48.60	8.35
5. Europe	3.18	1.07	3.09	0.83	2.63	0.80	1.34	0.33	51.50	11.45
**AUTHOR/UPLOADER**										
1. Academic institutions	3.92	0.86	3.84	0.80	2.92	0.64	1.49	0.22	55.50	4.90
2. Media	3.25	0.95	3.5	1.29	2.50	0.57	1.50	0.35	53.50	11.05
3. Health institutions	3.40	0.98	3.35	0.91	2.56	0.64	1.37	0.27	53.00	8.95
4. NGOs	3.28	0.48	3.14	0.69	2.28	0.95	1.25	0.26	48.55	10.65
5. Health individuals	3.47	0.76	3.15	0.78	2.42	0.75	1.33	0.28	51.00	10.65
6. Non-health individuals	2.11	0.60	2.11	0.33	1.88	0.33	1.09	0.35	45.55	4.60
**GENDER**										
1. Female	3.45	0.82	3.45	0.52	2.45	0.52	1.41	0.31	52.50	10.05
2. Male	3.36	0.93	3.20	0.93	2.48	0.73	1.33	0.29	51.50	9.30
**CANCER TYPE**										
1. Prostate	3.32	0.93	3.21	0.93	2.47	0.71	1.32	0.28	51.50	9.20
2. Uterus	4.50	0.70	4.00	0.00	3.50	0.70	1.59	0.10	62.50	3.50
3. Bladder	3.25	0.70	2.87	0.64	2.12	0.64	1.39	0.33	48.75	10.25
4. Colorectal	0.00	0.00	0.00	0.00	0.00	0.00	0.00	0.00	0.00	0.00
5. Other	4.00	0.70	3.80	0.44	2.80	0.44	1.56	0.33	58.00	9.05
**TOTAL**	**3.37**	**0.92**	**3.23**	**0.90**	**2.48**	**0.71**	**1.34**	**0.29**	**51.50**	**9.35**

**Table 3 healthcare-12-00243-t003:** Percentages of videos by the author.

Author/Uploader	%
1. Academic institutions	12.03%
2. Media	3.70%
3. Health institutions	34.25%
4. NGOs	6.48%
5. Health individuals	35.18%
6. Non-health individuals	8.33%

**Table 4 healthcare-12-00243-t004:** Percentages of videos by gender and cancer type.

Gender and Cancer Type	%
1. Female	10.19%
1. Prostate	0.00%
2. Uterus	4.63%
3. Bladder	0.93%
4. Colorectal	0.00%
5. Other	4.63%
2. Male	89.81%
1. Prostate	82.41%
2. Uterus	0.00%
3. Bladder	7.41%
4. Colorectal	0.00%
5. Other	0.00%

**Table 5 healthcare-12-00243-t005:** Pearson correlation coefficients and confidence intervals of quality scales.

	Pearson Correlation	95% Confidence Intervals (Bilateral) ^a^	
		Inferior	Superior
**DISCERN—GQS**	0.926 **	0.894	0.949
**DISCERN—JAMA**	0.772 **	0.682	0.838
**DISCERN—PEMAT**	0.747 **	0.650	0.820
**DISCERN—MQVET**	0.739 **	0.640	0.814
**GQS—JAMA**	0.812 **	0.737	0.868
**GQS—PEMAT**	0.791 **	0.708	0.853
**GQS—MQVET**	0.803 **	0.724	0.861
**JAMA—PEMAT**	0.697 **	0.585	0.783
**JAMA—MQVET**	0.812 **	0.736	0.867
**PEMAT—MQVET**	0.778 **	0.690	0.843

^a^. The estimation is based on Fisher’s transformation of R to Z. **. The correlation is significant at 0.01 level (bilateral).

**Table 6 healthcare-12-00243-t006:** Number of videos with high and low quality in each scale.

QUALITY SCALE	HQ	LQ
DISCERN	53	55
GQS	48	60
JAMA	50	58
PEMAT	59	49
MQ-VET	51	57
**TOTAL QUALITY**	**49**	**59**

**Table 7 healthcare-12-00243-t007:** Normalized scores of the dataset in the questionnaires.

Scale	Score Achieved	Maximum Possible Score	Normalized Score (Over 5.00)
DISCERN	3.37	5.00	3.37
GQS	3.23	5.00	3.23
JAMA	2.48	4.00	3.10
PEMAT	1.35	2.00	3.37
MQ-VET	51.90	75.00	3.46

## Data Availability

Data are available on request due to restrictions. The data presented in this study are available on request from the corresponding author.

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
