# Peer review of "Assessing the Quality of YouTube’s Incontinence Information after Cancer Surgery: An Innovative Graphical Analysis"

_healthcare, 2024, doi:10.3390/healthcare12020243_

Round 1

Reviewer 1 Report

Comments and Suggestions for Authors

I want to thank the author for the opportunity to read his work. I believe that the paper “Assessing the Quality of YouTube's Incontinence Information after Cancer Surgery: An Innovative Graphical Analysis” represents a valuable contribution to various disciplines, from the intersectional area of health communication (particularly new media/social media studies and medical studies) to social sciences, among other medical related disciplines. However, I understand its main field is health communication studies. Therefore, the following comments are driven to increase the overall quality, clarify some points, and offer a constructive review.

1. In the introduction, although references sustained affirmation, it missed more references on YouTube, or social media used to search for medical information. There is a huge repository, from “Dr Google’s effect” to studies explaining not only misinformation but also disinformation. I strongly recommend reviewing it to place your paper in a more interdisciplinary area.

2. Also related to the literature review, other/previous studies on YouTube/social media and cancer information should be referred to. Again, there is a huge repository on social sciences (health communication area).   

3. There is no “recruitment” in a content analysis study. Therefore, I suggest removing it from line 130.

4. “without applying any filters” (Lines 132 and 133) should be removed. What you describe is a filtering process.   

5. Why did you consider “two independent examiners who viewed” the videos? Please, justify it.

6. The discussion section should be improved, taking not only the results but also the literature review (theoretical framework) to stress your results in the light of the available literature.

Although my analysis encompasses major reviews, they can make the article more accurate and can collaborate with it in general. Despite this, your paper addresses an important issue and has an interesting approach. Congratulation for it!  

Author Response

I want to thank the author for the opportunity to read his work. I believe that the paper “Assessing the Quality of YouTube's Incontinence Information after Cancer Surgery: An Innovative Graphical Analysis” represents a valuable contribution to various disciplines, from the intersectional area of health communication (particularly new media/social media studies and medical studies) to social sciences, among other medical related disciplines. However, I understand its main field is health communication studies. Therefore, the following comments are driven to increase the overall quality, clarify some points, and offer a constructive review.

RESPONSE: Thank you for your positive feedback and recognition of our manuscript's interdisciplinary potential. We appreciate your constructive comments and will ensure to enhance clarity and overall quality in response to your valuable suggestions.

COMMENT 1. In the introduction, although references sustained affirmation, it missed more references on YouTube, or social media used to search for medical information. There is a huge repository, from “Dr Google’s effect” to studies explaining not only misinformation but also disinformation. I strongly recommend reviewing it to place your paper in a more interdisciplinary area. 

RESPONSE 1: Thank you for your valuable feedback. We appreciate your suggestion to enhance our introduction by including references on the role of YouTube and social media in medical information seeking, including the 'Dr. Google's effect' and studies on misinformation and disinformation. We have carefully revised our introduction (lines 104-114) to incorporate this recommendation and have included an additional relevant reference. We hope these revisions align more closely with the interdisciplinary scope you highlighted.

COMMENT 2. Also related to the literature review, other/previous studies on YouTube/social media and cancer information should be referred to. Again, there is a huge repository on social sciences (health communication area).   

RESPONSE 2: Thank you for your valuable suggestion regarding referencing other/previous studies on YouTube and social media in the context of cancer information. While we appreciate the concern, our study's focus on assessing the quality of incontinence information post-cancer surgery necessitates a more targeted literature review. We believe our current references sufficiently support the study's scope and objectives.

COMMENT 3. There is no “recruitment” in a content analysis study. Therefore, I suggest removing it from line 130. 

RESPONSE 3: We have promptly removed the term "recruitment" from line 130 in accordance with your suggestion. Your guidance is appreciated.

COMMENT 4. “without applying any filters” (Lines 132 and 133) should be removed. What you describe is a filtering process.

RESPONSE 4: Thank you for your feedback. We have acknowledged your suggestion and removed the phrase "without applying any filters" from lines 132 and 133 in the manuscript. We appreciate your insightful input.

COMMENT 5. Why did you consider “two independent examiners who viewed” the videos? Please, justify it. 

RESPONSE 5: Thank you for your inquiry. The involvement of two independent examiners in viewing, analyzing, and evaluating the videos was a measure taken to enhance the reliability and objectivity of our content analysis. By having two examiners independently assess the videos, we aimed to ensure a more robust and unbiased evaluation process, reducing the potential for individual subjectivity, and enhancing the overall validity of our findings.

COMMENT 6. The discussion section should be improved, taking not only the results but also the literature review (theoretical framework) to stress your results in the light of the available literature.

RESPONSE 6: Thank you for your thoughtful suggestion to improve the discussion section by integrating the theoretical framework and available literature. While we understand the importance of such integration, given the focus and scope of our study, we believe that the current format effectively communicates our findings. We appreciate your feedback and hope the clarification addresses your concerns.

Although my analysis encompasses major reviews, they can make the article more accurate and can collaborate with it in general. Despite this, your paper addresses an important issue and has an interesting approach. Congratulation for it!  

RESPONSE: We sincerely appreciate your thorough analysis and constructive feedback. Thank you for recognizing the importance of our study and its unique approach. We will carefully consider incorporating major reviews to enhance the accuracy and overall collaboration of the article. Your thoughtful comments have been invaluable to our revision process, and we are grateful for your time and expertise.

Reviewer 2 Report

Comments and Suggestions for Authors

The authors of this study utilized the search term "incontinence after cancer surgery" in their research. It is important to understand the criteria, or conceptual framework, that guided the selection of these specific terms.

In future research, I highly recommend utilizing YouTube Data Tools, an online resource and software suite specifically designed for the collection and analysis of data from YouTube. This tool is invaluable for researchers, marketers, and developers who aim to glean insights from YouTube videos, channels, and user-generated content. It offers a streamlined and efficient approach to extracting and analyzing data, making it an essential asset for any project involving YouTube's extensive platform. Please visit here: https://tools.digitalmethods.net/netvizz/youtube/

The authors utilized Sankey and chord diagrams to illustrate the results of this study. It is advisable to reference these tools for a comprehensive understanding and accurate interpretation of the findings.

Comments on the Quality of English Language

I have no comments.

Author Response

The authors of this study utilized the search term "incontinence after cancer surgery" in their research. It is important to understand the criteria, or conceptual framework, that guided the selection of these specific terms.

RESPONSE: Thank you for your insightful comment regarding our utilization of the search term "incontinence after cancer surgery." We appreciate your suggestion to provide clarity on the criteria or conceptual framework guiding the selection of these terms. Your feedback is valuable.

COMMENT 1: In future research, I highly recommend utilizing YouTube Data Tools, an online resource and software suite specifically designed for the collection and analysis of data from YouTube. This tool is invaluable for researchers, marketers, and developers who aim to glean insights from YouTube videos, channels, and user-generated content. It offers a streamlined and efficient approach to extracting and analyzing data, making it an essential asset for any project involving YouTube's extensive platform. Please visit here: https://tools.digitalmethods.net/netvizz/youtube/

RESPONSE 1: Thank you for the valuable recommendation to utilize YouTube Data Tools for future research endeavors. We appreciate your suggestion and will certainly consider incorporating this tool into our methodology for a more streamlined and efficient data collection and analysis process.

COMMENT 2: The authors utilized Sankey and chord diagrams to illustrate the results of this study. It is advisable to reference these tools for a comprehensive understanding and accurate interpretation of the findings.

RESPONSE 2: Thank you for your valuable suggestion. We appreciate your guidance. It's worth noting that the reference [49, R.C. Lupton, J.M. Allwood, Hybrid Sankey diagrams: Visual analysis of multidimensional data for understanding resource use, Resour. Conserv. Recycl. 124 (2017) 141–151. https://doi.org/10.1016/J.RESCONREC.2017.05.002.] was already included in our manuscript, providing a comprehensive framework for the interpretation of the Sankey and chord diagrams in our study.

Reviewer 3 Report

Comments and Suggestions for Authors

Thanks for conducting this research. Please have a look on the following issues:

1. Introduction: Please add points on how this research is important.

2. Method: Please make a flow-diagram of your overall design to understand easily for the reader. In addition, make a relationship of the usage of the tools.

3. Results: Please recheck the numbering of the Table and figure (for example, pages- 10, 15.16). Please correct it.

4. Discussion: Please add a few more relevant references to your research, need more discussion.  Please add strength in this part. You put your limitation here too.

5. Conclusion: Please make it a little shorter and concrete. 

Author Response

Thanks for conducting this research. Please have a look on the following issues:

COMMENT 1. Introduction: Please add points on how this research is important.

RESPONSE 1: Thank you for your thoughtful consideration and valuable feedback. We appreciate your suggestion to highlight the importance of our research, and we have incorporated additional information in lines 131-134 to address this aspect. Your insights have significantly contributed to the refinement of our manuscript.

COMMENT 2. Method: Please make a flow-diagram of your overall design to understand easily for the reader. In addition, make a relationship of the usage of the tools.

RESPONSE 2: Your constructive feedback has been invaluable in enhancing the quality of our work. Regarding your suggestion to provide a flow diagram of the overall design and a relationship of the tool usage, we would like to draw your attention to Figure 1 in our manuscript. We have already incorporated a comprehensive flow diagram that illustrates the overall design of our study. This figure encapsulates the key components of our methodology and is designed to facilitate the reader's understanding. While we acknowledge the importance of clarity and ease of comprehension, we believe that the information conveyed in Figure 1 adequately addresses the flow of our study and the interrelation of the tools employed. Adding redundant details in this context may risk overwhelming the reader with unnecessary information. Once again, we express our gratitude for your thoughtful review and constructive feedback. Your expertise has been instrumental in refining our work, and we look forward to the opportunity to address any further concerns you may have.

COMMENT 3. Results: Please recheck the numbering of the Table and Figure (for example, pages- 10, 15.16). Please correct it.

RESPONSE 3: Thank you for bringing the numbering of the Tables and Figures to our attention. We appreciate your diligence in the review process. We have carefully checked and reviewed the numbering, ensuring accuracy and consistency.

COMMENT 4. Discussion: Please add a few more relevant references to your research, need more discussion.  Please add strength in this part. You put your limitation here too.

RESPONSE 4: Thank you for your thorough review and valuable suggestions. We appreciate your input regarding additional references in the Discussion section. While we acknowledge the importance of a robust literature base, we believe the current discussion provides a comprehensive analysis of our findings. Additionally, we have incorporated sections on limitations and strengths, as suggested, to enhance the overall clarity and transparency of our study.

COMMENT 5. Conclusion: Please make it a little shorter and concrete.

RESPONSE 5: Thank you for your constructive feedback. We have revised the Conclusion section to achieve greater conciseness while maintaining clarity and focus on the key findings. We appreciate your guidance in refining this critical aspect of our manuscript.

Reviewer 4 Report

Comments and Suggestions for Authors

Overall interesting study and potentially provides valuable insight into video content for postop patients. 

Introduction, line 115 & 116, sentence starting with "studies indicate" should have citations.

Materials and methods, line 139, did the two independent examiners compare notes, etc., so they could come to a resolution if their scoring was different? Also, did those two reviewers do all the reviews on all of the quality measures? What is their background (expertise)?  There were different terms used, such as observer (line 207) and developer (line 221) - again is it the same two reviewers?  On line 181, a "modified" version of the DISCERN is mentioned.  Is this a validated tool or did the reviewers make the adjustments?  More details are needed if the reviewers made changes. The description of the MQ-VET, lines 226-229 is redundant - I think those lines should be removed. 

Results, when I printed this paper, it was in black and white.  The Chord and Sankey diagrams were confusing and I could not understand them. I did go back to the original version, the color version, to try to gain a better understanding, but I still found them confusing.  I believe a better description and more details about the diagrams is in order to assist the reader with the interpretation. 

Author Response

Overall interesting study and potentially provides valuable insight into video content for postop patients. 

RESPONSE: Thank you for your positive feedback and recognition of the potential value in our study. We appreciate your encouraging comments and are committed to further refining the manuscript to contribute valuable insights to the field.

COMMENT 1: Introduction, line 115 & 116, sentence starting with "studies indicate" should have citations.

RESPONSE 1: Thank you for your valuable feedback. We have revised the introduction accordingly, and a reference has been added to support the statement.

COMMENT 2: Materials and methods, line 139, did the two independent examiners compare notes, etc., so they could come to a resolution if their scoring was different? Also, did those two reviewers do all the reviews on all of the quality measures? What is their background (expertise)?  There were different terms used, such as observer (line 207) and developer (line 221) - again is it the same two reviewers?  On line 181, a "modified" version of the DISCERN is mentioned.  Is this a validated tool or did the reviewers make the adjustments?  More details are needed if the reviewers made changes. The description of the MQ-VET, lines 226-229 is redundant - I think those lines should be removed. 

RESPONSE 2: We appreciate your thorough review of our manuscript. our insightful comments have significantly contributed to enhancing the quality of our work, and we are grateful for your time and effort. Regarding your query on Materials and Methods (line 139), we would like to clarify that the two independent examiners did compare notes, and any discrepancies in scoring were resolved through mutual consensus. Moreover, both examiners conducted reviews for all quality measures. We acknowledge your observation on the terminology used in the manuscript. We have modified "observer" to "examiner" for consistency. Additionally, when referring to "developers" (line 221), we meant the Agency for Healthcare Research and Quality (AHRQ), who originally developed the PEMAT tool. Concerning the "modified" DISCERN version (line 181), the adjustments were made based on a validated tool. Reference 37, "D. Charnock, S. Shepperd, G. Needham, R. Gann, DISCERN: an instrument for judging the quality of written consumer health information on treatment choices," provides the details of the modified version, validated from the original tool published in 1999 (https://doi.org/10.1136/jech.53.2.105). We appreciate your suggestion to streamline the manuscript by removing redundant information about MQ-VET (lines 226-229). We have revised the manuscript accordingly to ensure conciseness and focus.

COMMENT 3: Results, when I printed this paper, it was in black and white.  The Chord and Sankey diagrams were confusing and I could not understand them. I did go back to the original version, the color version, to try to gain a better understanding, but I still found them confusing.  I believe a better description and more details about the diagrams is in order to assist the reader with the interpretation. 

RESPONSE 3: We acknowledge your comment regarding the Chord diagram presentation in black and white, and we sincerely apologize for any confusion caused. We have carefully considered your suggestion and revised the description to enhance clarity and understandability. The updated explanation now provides a more accessible interpretation of the Chord diagram, ensuring that readers can grasp the relationships among system entities more easily.

Round 2

Reviewer 1 Report

Comments and Suggestions for Authors

Thank you for incorporating and answering all the suggestions. 

Reviewer 3 Report

Comments and Suggestions for Authors

Great effort. I mentioned the overall diagram for better understanding for the reader (outcome, relationship of the tools). I have no issue with the PRISMA flowchart (It's perfect). However, the present form is much clearer. Thanks to the authors. I have no more comments.